# Genetic Approach to Mitigate Hallucination in Generative IR

Hrishikesh Kulkarni
first@ir.cs.georgetown.edu
Georgetown University
Washington, DC, USA

Nazli Goharian
first@ir.cs.georgetown.edu
Georgetown University
Washington, DC, USA

Ophir Frieder
first@ir.cs.georgetown.edu
Georgetown University
Washington, DC, USA

Sean MacAvaney
first.last@glasgow.ac.uk
University of Glasgow
Glasgow, UK

## ABSTRACT

Generative language models hallucinate. That is, at times, they generate factually flawed responses. These inaccuracies are particularly insidious because the responses are fluent and well-articulated. We focus on the task of Grounded Answer Generation (part of Generative IR), which aims to produce direct answers to a user's question based on results retrieved from a search engine. We address hallucination by adapting an existing genetic generation approach with a new 'balanced fitness function' consisting of a cross-encoder model for relevance and an n-gram overlap metric to promote grounding. Our balanced fitness function approach quadruples the grounded answer generation accuracy while maintaining high relevance.

## 1 INTRODUCTION AND RELATED WORK

Grounded answer-generation approaches generate answers based on top retrieved results to the user queries. Although producing highly relevant responses, they still suffer from hallucination. To address this issue, we model Generative Information Retrieval as a Genetic Algorithm with a fitness function based on a simple-yet-effective n-gram overlap metric. This results in relevant and consistent output, namely lowering the frequency of hallucination. We evaluated[1] our method "Genetic Approach using Grounded Evolution" (GAuGE) across three datasets using four different models to demonstrate effectiveness and utility. We found that it reduces hallucination without impacting the relevance of generated answers. Our main contributions are as follows:

- **Relevance:** GAuGE maintains high relevance to the query.
- **Comprehensiveness:** GAuGE provides more comprehensive answers as multiple seed documents are taken into consideration.
- **Groundedness:** Most importantly, GAuGE produces factual results with minimal hallucination.

Generative models target the generation of answers, overcoming typical search space boundaries by interacting in parametric space [13, 16]. These large language models (LLMs) support applications ranging from summarization to conversational search but are hindered by hallucination. Efforts to mitigate hallucinations vary; they address hallucination at differing stages of the computation, namely in pre-, during, and post-generation.

**Pre-generation approaches** focus on improvised training and tuning. They either look for comprehensive correction or improvement across the training cycle by weight adjustments. They work at the model level using external custom data for model enhancement and fine-tuning [26]. Methodologies belonging to this approach are: learning from human feedback [11], direct optimization with human feedback (Chain-of-Hindsight [21]), reward modeling with reinforcement learning (RLHF [25]) and learning with automated feedback [26].

**During-generation approaches** use re-ranking and feedback-guided approaches. Effective re-ranking methods use a neural transformer model [12] and weighted voting scheme to filter incorrect answers [19]. During-generation models typically use Retrieval Augmented Generation (RAG) models [5, 17] where context / top retrieved results are given as input to reduce hallucination. Chain of Verification (CoVe) generates a series of questions to verify factuality [9]. The Decoding by Contrasting Layers (DoLa) model mitigates hallucinations by amplifying the factual information in the mature layer and understating the linguistic predominance in the premature layer [6]. There are also additional approaches namely self-correction, correction with external feedback and multi-agent debate [26]. A self-correction framework usually has a single pretrained framework as proposed in the Self-Refine model [22].

**Post-generation correction approaches** are applied after complete response generation. Some external fact-checking and verification modules were used to detect hallucinated content at the post-generation stage [27].

Efforts have been taken across multiple stages in specific domains like finance to deal with hallucination [31]. The proposed method, GAuGE, is a during-generation correction method with the key difference being the genetic modeling with a cross-encoder and n-gram overlap based fitness function.

**Evaluations** are typically carried out on objective datasets where the list of probable answer entities is known [9, 29]. A few efforts including HaluEval also perform manual annotation for hallucinated content detection [10, 18, 29]. Recent efforts like DelucionQA show that automatic evaluation of hallucination is important [32]. Our primary focus is on the mitigating it in descriptive short answers. Fact verification model based metrics can be used for hallucinated content detection in this setup as shown in FActScore, where automatic metrics strongly align with human annotations [23].

---

[1]Complete implementation: https://github.com/Georgetown-IR-Lab/GAuGE.

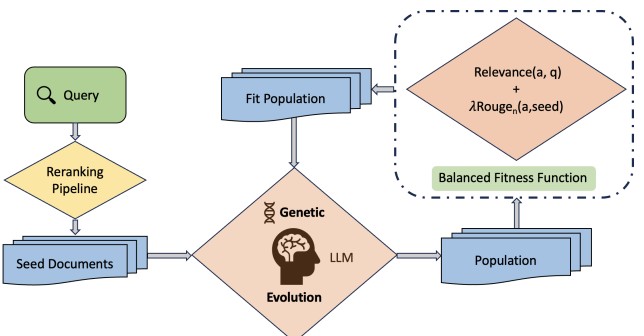

**Figure 1: System Architecture. Answer: a, Query: q, Seed documents: seed.**

Hence, we use a set of claim verification models [33] which sustain high accuracy on fact verification benchmarks and are suitable for descriptive short answers. The literature underlines the need for factual answer generation models that limit hallucination.

## 2 PROPOSED METHOD: GAUGE

We first generally describe genetic generative approaches then detail our implementation GAᴜGE.

### 2.1 Genetic Generative Approaches

Genetic generative approaches use generative language models as a genetic operator [14]. GAᴜGE has three genetic operators with respective prompts:

- Randomized operator: Random mutation or rewrite the document. Prompt: 'Summarize the document'.
- Controlled mutation: Query specific document rewriting. Prompt: 'Re-write the document to better answer the query'.
- Cross-over: Two or more document-based rewriting to generate single query specific answer. Prompt: 'Re-write the given documents to better answer the query'.

The system architecture of GAᴜGE is depicted in Figure 1. In this setup, initially retrieved documents are referred to as seed population. The initial retrieval is performed using a multi-stage pipeline with lexical method for first-stage retrieval and a cross-encoder model for re-ranking. The fit documents from the previous iteration survive to participate in the next iteration. Here, the grounded fitness function determines the fit population. The weighted combination of being relevant and grounded determines the fitness score. The evolution continues until the exit criteria is met. No new entry or rank change in top $d$ documents flags the termination.

### 2.2 GAᴜGE

We detail GAᴜGE in Algorithm 1. Rather than making a single pass over the results, GAᴜGE iteratively uses genetic operators for controlled mutations and crossovers along with a grounded fitness function. Here LLM based mutation and random operator counter the local maxima problem.

Our initial retrieval results are our seed documents. Document text is used as the genetic representation. The generative language model is the genetic operator which performs 'mutations' and 'crossovers' using specifically designed prompts. A grounded fitness

---

**Algorithm 1** GAᴜGE

**Input:** $q$ query, $D_0$ document corpus
**Output:** $d$ relevant, comprehensive and grounded answer
   $seeds \leftarrow \textsc{ReRanker}(\textsc{FirstStage}(q, D_0))$   ▷ seed candidates
   $D \leftarrow \textsc{GeneticOperators}(seeds)$   ▷ generate new candidates
   $D \leftarrow \textsc{Relevance}(q, D) + \lambda \times \textsc{Rouge}_N\text{F1}(D, seeds)$   ▷ fit candidates
   **while** Termination Criteria **do**
      $D \leftarrow \textsc{GeneticOperators}(D)$   ▷ generate new candidates
      $D \leftarrow \textsc{Relevance}(q, D) + \lambda \times \textsc{Rouge}_N\text{F1}(D, seeds)$  ▷ fit candidates
   **end while**

---

function $f$ (defined in Equation 1) decides survivors for the next iteration in the evolutionary cycle. It prioritizes factual outcomes.

$$f = Relevance(query, D) + \lambda \times Rouge_n F1(D, seeds) \qquad (1)$$

Here $D$ is the set of documents generated after invoking the generative language model; seeds are the seed documents, and $\lambda$ is the scaling parameter. The objective of the Relevance function is to produce answers of greater relevance to the query while that of the rouge metric is to ensure grounding of the generated answer in the seed documents. Thus, mutations and cross-overs performed by the LLM rewriter try to generate answers closer to user information need. The grounded fitness function ensures that among generated outputs, relevant and grounded answers get precedence over other answers in generating new off-springs. This ensures a balance between escaping from the perceived boundaries of the traditional retrieval system and producing both factual and relevant answers.

## 3 EXPERIMENT

We investigate and answer the following research questions:

**RQ1:** Does an n-gram overlap metric based fitness function achieve a consistent and measurable drop in hallucination?
**RQ2:** With GAᴜGE, is there a tradeoff between relevance retention and hallucination mitigation?
**RQ3:** Is GAᴜGE robust across generative language models used?
**RQ4:** How does change in the type of Rouge metric impacts hallucination and relevance?

### 3.1 Datasets

The studied language in this work is English (BenderRule [3]). To evaluate GAᴜGE effectiveness, we use the following datasets:

- **MS Marco Dev (small) - Dev(sample).** First 100 queries (sorted by id) from the Dev (small) subset used for evaluation [2].
- **TREC 2019 Deep Learning (Passage Subtask) - DL 19** This dataset contains 43 queries along with manual judgements [8].
- **TREC 2020 Deep Learning (Passage Subtask) - DL 20.** This dataset contains 54 queries along with manual judgements [7].

### 3.2 Models and Baselines

In GAᴜGE, the initial seed population is determined through retrieval using a multi-stage pipeline where we use BM25 [30] for first stage retrieval and the cross-encoder model Electra [28] for re-ranking. Use of a cross-encoder model ensures quality of seed population. We use GPT-3 [4] `text-edit-davinci-001` and GPT-4 [1] with the default parameters as LLM rewriters to perform query-specific genetic operations: 'mutations' and 'crossovers'. Our fitness function comprises of two main components to balance grounding

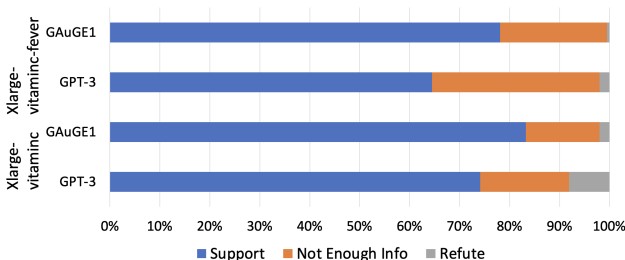

**Figure 2: Comparison between GPT-3 and GAᴜGE in mitigating hallucinations across datasets**

| Model | Dev (sample) | | | DL'19 | | | DL'20 | | |
|---|---|---|---|---|---|---|---|---|---|
| | + | = | − | + | = | − | + | = | − |
| GPT-3 | 36 | 8 | 56 | 15 | 4 | 24 | 22 | 3 | 29 |
| Gen²IR | 88 | 0 | 12 | 40 | 0 | 3 | 47 | 0 | 7 |
| GAᴜGE1 | 80 | 1 | 19 | 35 | 0 | 8 | 45 | 1 | 8 |
| GAᴜGE2 | 63 | 6 | 31 | 30 | 0 | 13 | 40 | 2 | 12 |

**Table 1: Relevance as per MonoT5: '+' denotes cases where model output is preferred; '−' denotes cases where Re-ranking output is preferred; '=' denotes cases where both outputs are equivalent. GAᴜGE1 uses Rouge1, GAᴜGE2 uses Rouge2.**

and relevance. The first component uses the cross-encoder model Electra [28] that takes the query and candidate answer into account. The second uses the Rouge metric [20] to measure n-gram overlap between the candidate answers and the seed document. We experimented with Rouge1, Rouge2 and RougeL. GAᴜGE1 uses Rouge1 while GAᴜGE2 uses Rouge2. $\lambda$ parameter acts as the scaling factor between Rouge metric and relevance from cross-encoder model. In our experiments we give equal importance to both of them.

As shown in Figure 2, we compare GAᴜGE to GPT-3 [4] by providing the query along with seed documents as input and prompting to generate a query-specific answer. We also compare GAᴜGE with Gen²IR method [14] which is another approach built on top of GPT trying to model generative Information Retrieval as a genetic algorithm with focus on relevance. We reproduce results of Gen²IR using the parameters recommended for the genetic process [14] and use the same for our proposed model evaluation. Here, we generate twelve offsprings from top two documents in each iteration.

## 3.3 Evaluation

We use the **MonoT5** cross-encoder model [24] to evaluate answer relevance to the query. We use the **ALBERT-base** and **ALBERT-xlarge** models [15] fine-tuned on challenging contrastive fact verification dataset **VitaminC** [33] and fact verification benchmark dataset **FEVER** [34] to assess presence of hallucinated content in the generated answers. The above models show 86% to 96% accuracy on fact verification benchmarks like FEVER and VitaminC and are highly reliable [33]. This set of fact verification models classify a generated answer into three categories namely: 'SUPPORTS', 'REFUTES' and 'NOT ENOUGH INFO'. The class is SUPPORTS when the generated answer is completely supported by the contents of the seed documents. The class is REFUTES when it contradicts the

contents of the seed documents. The class is NOT ENOUGH INFO when the claims in the generated answer are neither supported by nor contradict the contents of the seed documents. Our primary interest lies in the percentage of grounded answers, i.e., completely supported by contents of seed documents with no hallucination. We acknowledge limitations of auto evaluations and try to mitigate these by using different models in the algorithm and evaluations.

## 4 RESULTS AND ANALYSIS

We now address our four research questions.

### 4.1 RQ1: Grounded - No Hallucination

We first compare GAᴜGE with GPT-3. As evident in Table 1, relevance of generated answers by directly invoking GPT-3 is on the lower side, and hence, it is not considered as a primary baseline. When evaluated, GAᴜGE1 outperforms GPT-3 in generating grounded answers as evident in Figure 2.

We evaluated GAᴜGE with variants of both ALBERT-base and ALBERT-xlarge models. Across evaluation models, GAᴜGE2 is better at mitigating hallucinations than GAᴜGE1 with Gen²IR as our baseline, see Table 2 and 3. When evaluated with the ALBERT-base-vitaminc fact verification model, the accuracy of grounded answer generation increases from 0.023 to 0.605 for DL 19, from 0.148 to 0.648 for DL 20 and from 0.110 to 0.560 for Dev (sample) dataset. Similarly, when evaluated using ALBERT-base-vitaminc-fever fact verification model which is finetuned on the FEVER benchmark dataset, the accuracy increases from 0.023 to 0.512 for DL 19, from 0.111 to 0.667 for DL 20 and from 0.110 to 0.590 for Dev (sample).

ALBERT-xlarge models are the best performing models for the claim verification task [33]. As evident in Table 2 and 3, we also evaluate using ALBERT-xlarge-vitaminc and ALBERT-xlarge-vitaminc-fever with Gen²IR as a baseline. Here, in case of GAᴜGE1, the accuracy of grounded answer generation increases from 0.186 to 0.930 for DL 19, from 0.204 to 0.834 for DL 20 and from 0.250 to 0.790 for Dev (sample) dataset. Further, accuracy increases from 0.186 to 0.930 for DL 19, from 0.204 to 0.834 for DL 20 and from 0.250 to 0.790 for Dev (sample) dataset for the respective evaluation model. Similar imporvements are also observed in case of GPT-4. Overall, we conclude that the accuracy has at least quadrupled across four evaluation models, and GAᴜGE is highly effective at generating grounded answers and mitigating hallucinations addressing RQ1.

### 4.2 RQ2: Maintaining High Relevance

We primarily discuss GAᴜGE1 results from Table 1 as it leads to highest relevance among GAᴜGE variants. As evident in Table 1, the number of queries where GAᴜGE1 is preferred drops by 5 for DL 19, by 2 for DL 20 and by 8 for Dev (sample). For reference, there are 43 queries in DL 19, 54 queries in DL 20 and 100 queries in Dev (sample) dataset. Hence, out of 197 queries we observe a drop in relevance for 15 queries. On the other hand, the percentage of grounded answers quadrupled by using GAᴜGE. Similar trend is observed when using GAᴜGE1 with GPT-4 where there is a drop in relevance by 3 on DL 19 dataset while doubling the percentage of grounded answers. Hence, we infer that GAᴜGE drastically reduces hallucination while maintaining relatively high relevance results addressing RQ2.

| LLM | Method | Sup | NEI | Ref | Acc. | Sup | NEI | Ref | Acc. | Sup | NEI | Ref | Acc. | Sup | NEI | Ref | Acc. |
|---|---|---|---|---|---|---|---|---|---|---|---|---|---|---|---|---|---|
| GPT-3 | Gen$^2$IR | 1 | 40 | 2 | 0.023 | 1 | 42 | 0 | 0.023 | 8 | 34 | 1 | 0.186 | 10 | 33 | 0 | 0.233 |
| | GAᴜGE1 | 26 | 17 | 0 | 0.605 | 22 | 20 | 1 | 0.512 | 40 | 3 | 0 | 0.930 | 31 | 12 | 0 | 0.721 |
| | GAᴜGE2 | 37 | 6 | 0 | 0.861 | 37 | 6 | 0 | 0.861 | 41 | 2 | 0 | 0.954 | 36 | 7 | 0 | 0.837 |
| GPT-4 | Gen$^2$IR | 16 | 26 | 1 | 0.372 | 17 | 26 | 0 | 0.395 | 21 | 21 | 1 | 0.488 | 23 | 20 | 0 | 0.535 |
| | GAᴜGE1 | 29 | 13 | 1 | 0.674 | 33 | 10 | 0 | 0.767 | 39 | 3 | 1 | 0.907 | 39 | 4 | 0 | 0.907 |
| | GAᴜGE2 | 36 | 7 | 0 | 0.837 | 41 | 2 | 0 | 0.954 | 40 | 3 | 0 | 0.930 | 41 | 2 | 0 | 0.954 |

**Table 2: Hallucination: evaluated by ALBERT models: base-vitaminc, base-vitaminc-fever, xlarge-vitaminc and xlarge-vitaminc-fever respectively. MSMARCO TREC DL 19 dataset used. GAᴜGE1 uses Rouge1, GAᴜGE2 uses Rouge2. Sup: Support, NEI: Not Enough Info, Ref: Refutes, Acc: Accuracy.**

| Dataset | Method | Sup | NEI | Ref | Acc. | Sup | NEI | Ref | Acc. | Sup | NEI | Ref | Acc. | Sup | NEI | Ref | Acc. |
|---|---|---|---|---|---|---|---|---|---|---|---|---|---|---|---|---|---|
| DL 20 | Gen$^2$IR | 8 | 43 | 3 | 0.148 | 6 | 45 | 3 | 0.111 | 11 | 33 | 10 | 0.204 | 12 | 42 | 0 | 0.222 |
| | GAᴜGE1 | 35 | 18 | 1 | 0.648 | 36 | 18 | 0 | 0.667 | 45 | 8 | 1 | 0.834 | 45 | 9 | 0 | 0.833 |
| | GAᴜGE2 | 46 | 8 | 0 | 0.852 | 48 | 6 | 0 | 0.889 | 51 | 3 | 0 | 0.944 | 50 | 4 | 0 | 0.926 |
| Dev (sample) | Gen$^2$IR | 11 | 82 | 7 | 0.110 | 11 | 84 | 5 | 0.110 | 25 | 65 | 10 | 0.250 | 22 | 76 | 2 | 0.220 |
| | GAᴜGE1 | 56 | 43 | 1 | 0.560 | 59 | 41 | 0 | 0.590 | 79 | 18 | 3 | 0.790 | 78 | 21 | 1 | 0.780 |
| | GAᴜGE2 | 83 | 17 | 0 | 0.830 | 81 | 19 | 0 | 0.810 | 91 | 9 | 0 | 0.910 | 89 | 11 | 0 | 0.890 |

**Table 3: Hallucination: evaluated by ALBERT models: base-vitaminc, base-vitaminc-fever, xlarge-vitaminc and xlarge-vitaminc-fever respectively. GAᴜGE1 uses Rouge1, GAᴜGE2 uses Rouge2. All methods use GPT-3. Sup: Support, NEI: Not Enough Info, Ref: Refutes, Acc: Accuracy.**

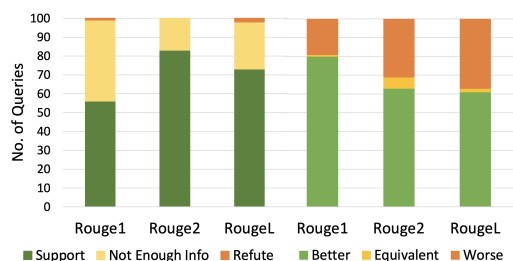

**Figure 3: Hallucination mitigation and Relevance with GPT-3 based GAᴜGE using three Rouge metrics. Relevance comparisons are with respect to top retrieved results.**

### 4.3 RQ3: Robustness across different LLMs

As evident in Table 2, we evaluated both GPT-3 and GPT-4 based GAᴜGE and Gen$^2$IR. GPT-4 is more advanced LLM than GPT-3 [1] and hence in most cases, GAᴜGE shows better effectiveness at generating grounded answers and mitigating hallucinations with GPT-4 as compared to GPT-3 addressing RQ3.

### 4.4 RQ4: Different Rouge Metrics

We experimented with different Rouge metrics, namely Rouge1, Rouge2 and RougeL, in the fitness function to evaluate changes in grounded answer generation and relevance to the user query. We perform these experiments on the first 100 queries from the Dev (small) dataset. As evident in Figure 3, Rouge2 is most effective in mitigating hallucinations followed by RougeL and Rouge1. Rouge2 generates grounded answers in 83% cases while RougeL and Rouge1 generate grounded answers in 73% and 56% cases respectively. But the performance of Rouge2 and RougeL comes at

the cost of relevance. As evident in Figure 3, Rouge1 generates most relevant answers to the user query. Rouge1 generates answers more relevant than the top retrieved result in 80% cases while Rouge2 and RougeL do so in 63% and 61% cases respectively. Use of Rouge1 in fitness function mitigates hallucination while maintaining high relevance. On the other hand, Rouge2 is more effective than Rouge1 at mitigating hallucinations but it also results in reduced relevance. Hence, as evident in Tables 1, 2 and 3 we use both Rouge1 and Rouge2 for extensive evaluation across datasets. In summary, applications favoring relevance, should use Rouge1 while applications favoring grounded answer generation should use Rouge2 to address RQ4.

## 5 ERROR ANALYSIS

We analyzed and will address in future work the cases in which GAᴜGE fails to mitigate hallucinations. Based on the patterns observed, we classify cases of failure into three types namely: Presence of Numbers and Math, Difficult Vocabulary (e.g. complex medical terms) and Ambiguous Queries. The occurrence of these cases are query specific but mostly dominated by difficult vocabulary.

## 6 CONCLUSION

GAᴜGE introduces the combination of grounded, genetic and generative methodologies through a balanced fitness function. It strikes the balance between relevance and factuality. GAᴜGE delivers significant improvements in mitigating hallucination while retaining relevance to the user query. The described approach furthers the possibility to eventually rely on generative models in critical and real time applications where there is minimal tolerance for hallucination. As future work, we plan to address the limitations stated in the error analysis section. Further, time complexity aspect of this algorithm can be looked at critically along with exploration beyond the obvious applications.

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
