# OpenReview forum: "Genetic Approach to Mitigate Hallucination in Generative IR"
_ACM.org/SIGIR/2024/Workshop/Gen-IR — Gen-IR_SIGIR24_

### Official Review · Reviewer_fzLq · 2024-05-24

**Rating:** 1
**Confidence:** 3

**Review:**

### Summary
The paper proposes GAuGE, a genetic approach that uses a balanced fitness function consisting of a cross-encoder model for relevance and an n-gram overlap metric to promote grounding. With such approach, it shows better ability on generating response by grounding on the context, mitigating hallucination.

### Strength
1. The paper is clearly written.
2. GAuGE, though simple, significantly improves grounded answer generation accuracy while maintaining high relevance.
3. The paper shows experiments over various LLMs showing the generalizability.

### Weakness
1. More analysis of GAuGE would help in understanding benefits of the approach.
2. Comparison with other approaches that try to mitigate hallucination would be necessary such as [1].
3. To investigate whether it truly mitigates hallucination, adding results over counterfactual [2] or new knowledge [3] dataset would be interesting.

[1] Trusting Your Evidence: Hallucinate Less with Context-aware Decoding

[2] Adaptive Chameleon or Stubborn Sloth: Revealing the Behavior of Large Language Models in Knowledge Conflicts

[3] ALCUNA: Large Language Models Meet New Knowledge

---

### Official Review · Reviewer_Qr6a · 2024-05-24
**The unique contributions of this paper need to be highlighted more clearly.**

**Rating:** -1
**Confidence:** 3

**Review:**

This paper proposes a genetic algorithm to mitigate hallucination of generative language models, hence relevant, comprehensive and grounded answer could be generated.
Strengths:
* This paper is well-structured.
* The experiments in this paper are well done and the experimental analysis is exhaustive.
* Integrating genetic algotithms and prompt-based methods sounds novel and exciting.

Weaknesses:
* The differences between GAUGE and GEN2IR aren't clearly clarified. Actually the motivation, contributions and article structure of both papers seem close, please highlight the unique contributions of this paper.
* The methodology proposed in the Section 2 does not seem to have been clearly articulated.
* This paper focuses on the answer generation of generative language models, while the IR stage remains lexical method for first-stage retrieval and a cross-encoder model for re-ranking.  To this end, I don't think the paper fits the "generative IR" topic of this workshop well.

---

### Decision · Program_Chairs · 2024-05-31

**Decision:**

Accept

**Comment:**

The paper proposes GAuGE, a genetic approach that uses a balanced fitness function consisting of a cross-encoder model for relevance and an n-gram overlap metric to promote grounding. The chairs consider that GAuGE and Gen2IR have enough differences in their methodology.

As noted by the reviewers, the chairs would like to point out that using a black-box LLM in an iterative algorithm comes with scalability challenges, and the authors used fewer than 200 queries for evaluation. Additionally, it would be desirable to benchmark this methodology against other hallucination mitigation methods that are not genetic. Furthermore, the methodology section needs to be clearer. We hope that the authors can address these points in the camera-ready version.